# Soluble Collectin 11 (CL-11) Acts as an Immunosuppressive Molecule Potentially Used by Stem Cell-Derived Retinal Epithelial Cells to Modulate T Cell Response

**DOI:** 10.3390/cells12131805

**Published:** 2023-07-07

**Authors:** Giorgia Fanelli, Marco Romano, Giovanna Lombardi, Steven H. Sacks

**Affiliations:** Peter Gorer Department of Immunobiology, School of Immunology and Microbial Sciences, King’s College, London SE1 9RT, UK; marco.romano@kcl.ac.uk (M.R.); giovanna.lombardi@kcl.ac.uk (G.L.); steven.sacks@kcl.ac.uk (S.H.S.)

**Keywords:** T cell activation, stem cell-derived retinal pigment epithelium cells, Collectin 11, transplantation

## Abstract

Retinal pigment epithelium (RPE) cell allotransplantation is seen as a possible solution to retinal diseases. However, the RPE-complement system triggered by the binding of collectin-11 (CL-11) is a potential barrier for RPE transplantation as the complement-mediated inflammatory response may promote T cell recognition. To address this, we investigated the role of CL-11 on T cell immuno-response. We confirmed that RPE cells up-regulated MHC class I and expressed MHC class II molecules in an inflammatory setting. Co-cultures of RPE cells with T cells led to the inhibition of T cell proliferation. We found that CL-11 was partially responsible for this effect as T cell binding of CL-11 inhibited T cell proliferation in association with the downregulation of CD28. We also found that the suppressive action of CL-11 was abrogated in the presence of the RGD peptide given to block the T cell binding of CL-11 by its collagen-like domain. Because RPE cells can bind and secrete CL-11 under stress conditions, we postulate that soluble CL-11 contributes to the immunosuppressive properties of RPE cells. The investigation of this dual biological activity of CL-11, namely as a trigger of the complement cascade and a modulator of T cell responses, may provide additional clues about the mechanisms that orchestrate the immunogenic properties of RPE cells.

## 1. Introduction

Visual impairment can often result from ocular inflammation. It is now well known that ocular diseases such as age-related macular degeneration (AMD), uveitis, and diabetic retinopathy (DR) have a strong immunological contribution [1,2,3]. The definition of the eye as an immune-privileged site has evolved from a concept of a passive physical barrier to a dynamic process in which intraocular inflammation is modulated by regulatory molecules and cells in the eye [4,5]. The retinal pigment epithelium (RPE) is a polarized pigmented monolayer of epithelial cells situated between the retinal photoreceptors and choroid, which performs several functions that are essential for the survival and function of the photoreceptor cells [6,7]. Therefore, damage to the RPE is linked to irreversible sight loss.

Stem cell-derived RPE transplantation has been explored as a potential treatment of retinal diseases that, if achieved, could restore sight. While the safety and efficiency of the cell product have been extensively explored, the immunological properties of stem cell-derived RPE cells are still under review. There is uncertainty about the antigenicity of RPE cells because of the contradictory immunosuppressive and immunogenic roles of RPE cells under inflammatory conditions. In vitro, RPE cells suppress the allogeneic T cell response to inflammation through the expression of RPE cell surface molecules and soluble inhibitory factors such as programmed cell death 1 (PD-L1) [8], cytotoxic T lymphocyte-associated antigen 2 alpha (CTLA-2α) [9], retinoic acid [10], and transforming growth factor beta (TGF-β) [11]. However, especially upon IFN-γ exposure, RPE cells can change their properties to immunogenic. They up-regulate MHC-I and PD-L1 molecules and express MHC-II [12,13] to become capable of inducing an immune response. In addition, RPE cells secrete cytokines (IL-6) [14] and chemokines (CCL2, CXCL9, 10, and 11) which promote the recruitment of inflammatory cells [15].

Also relevant is a recent appreciation for the role of the complement in the pathology of RPE in retinal diseases. The complement cascade is a crucial effector mechanism of the innate immune system contributing to the clearance of pathogens as well as participating in the extent of the inflammatory immune response [16]. RPE cells are a prominent source of complement proteins in the eye, including core components required for complement activation (C3 C5 and Factor b (Fb) [17,18], and complement control proteins (CD46, complement factor H (CFH), CD59, and complement factor I (CFI)) [19,20,21] required to terminate the activation of the complement system and restore the retinal immune homeostasis. A strong line of evidence has shown the involvement of complement activation in early and late forms of AMD. The pathology of AMD is strongly associated with SNP variation in several complement components, including CFH, C3, CFB, and C9 [22,23]. This has led to the idea that dysregulation of complement activation is a critical driver in the pathophysiology of AMD.

We have previously shown that under hypoxic conditions, stem cell-derived RPE cells secrete complement components, including collectin 11 (CL-11) [24]. CL-11, also known as collectin kidney 1 (CL-K1), is a soluble C-type lectin composed of a globular head containing a carbohydrate recognition domain (CRD) followed by a neck region and a collagen-like domain (CLD). The CRD domain of CL-11 is responsible for binding, in a Ca2+-dependent manner, to a broad range of pathogens [25,26,27] and to hypoxia-stressed epithelial cells, typically leading to complement activation [24,28,29]. The binding of the CLD of CL-11 to other structures expressed on mammalian cells, e.g., calreticulin (CRT), mediated a non-complement function such as phagocytosis [30] suggesting that CL-11 can serve different biological functions depending on the mechanism of binding through the CRD or the CLD.

Having shown that stressed RPE can initiate complement cascade through the binding of CL-11, here we investigated its direct effect on CD4+CD25− T cells, which mediate the cognate immune response to alloantigen. We found that the binding of CL-11 to the T cell surface inhibited T cell proliferation, and this correlated with a dramatic downregulation of CD25 and CD28 and, to a lesser extent, HLA-DR. Importantly, the binding of CL-11 to the T cell surface appeared to occur via its CLD, opening the question of whether a complement-independent role of CL-11 on immune cells could be beneficial to down-modulate the T cell response following RPE transplantation.

## 2. Materials and Methods

### 2.1. RPE Cell Culture and Phenotype

Thawed human retinal pigment epithelium cells (Lonza, Basel, Switzerland) were seeded at the density of 10^4^ cells/cm^2^ (according to Lonza’s instructions) and maintained in Retinal Pigment Epithelial Cell Basal Medium (Lonza, Basel, Switzerland), containing supplements (L-glutamine, GA-1000, and bFGF (Lonza, Basel, Switzerland) and 2% FBS (Lonza, Basel, Switzerland) until 90–95% confluent. Medium was then changed to retinal differentiation medium (RDM) (DMEM, F12, Pen/Strep, and B27 without retinoic acid). Lightly pigmented areas of RPE appeared as early as week three in culture, and RPE cell cultures were fed every two days. For some experiments, RPE cells were treated with 500 IU/mL of recombinant hIFN-γ (rhIFN- γ, R&D System, Minneapolis, MN, USA) for three days or under hypoxic conditions for 24 h as previously described [24,31]. For detection of cell surface markers, RPE cells were treated with TrypLE Express enzyme solution (Thermo Fisher Scientific, Waltham, MA, USA) for 10 min, and single-cell suspensions were stained with LIVE/DEAD™ Fixable Near-IR Dead Cell Stain Kit according to the manufacturer’s instructions (Thermo Fisher Scientific, Waltham, MA, USA). Cells were then washed and labeled with a combination of antibodies to surface molecules: HLA-ABC, PD-L1, HLA-DR, CD80, CD86, and CD140b. Stained cells were fixed with 4% PFA (paraformaldehyde, Sigma-Aldrich, St. Louis, MO, USA) and acquired using LSRFortessa X20 (BD Bioscience, Franklin Lakes, NJ, USA). All flow cytometry data were analyzed using FlowJo software (version 10.8.51 for Mac). The list of all labeled antibodies used in this study can be found in Appendix A.

### 2.2. Cell Isolation

Peripheral blood was obtained from anonymized leukocyte cones supplied by the National Blood Transfusion Service (NHS Blood and Transplantation, Tooting, London, UK). Peripheral blood mononuclear cells (PBMC) were isolated by lympholyte (1.077 g/cm^3^) gradient stratification (Lymphoprep, Axis-Shield, Oslo, Norway). RosetteSep Human CD4+ T cell enrichment cocktail (STEMCELL Technologies, Vancouver, BC, Canada) was used to purify CD4+ T cell fraction, and CD25 Microbeads II (Miltenyi Biotec, Surrey, UK) were used to separate CD4+CD25− T cells from CD4+CD25+ T cell fraction according to manufacturer’s instructions.

### 2.3. Co-Culture Assay

5 × 10^5^ PBMCs or CD4+CD25− T cells were labeled with 2.5 μM of Carboxyfluorescein succinimidyl ester (CFSE, Thermo Fisher Scientific, Waltham, MA, USA) for 4 min at room temperature, stimulated or not with 0.5 µg/mL soluble anti-CD3 Ab (clone OKT3, Cat. 16003785, eBioscience, San Diego, CA, USA) and co-cultured for four days in RMPI 10% heat-inactivated FBS (Thermo Fisher Scientific, Waltham, MA, USA) with or without adherent RPE cells (ratio 1:1) (48-well plate, VWR, Leicestershire, UK). For some experiments, RPE cells were pre-treated with rhIFN-γ, hypoxia, or co-culture in the presence of 10^4^ allogeneic dendritic cells (DCs) generated as previously described [32]. Cell proliferation was determined by monitoring CFSE dilution.

### 2.4. Cytokine Intracellular Staining

Following co-culture with RPE cells, PBMCs were stimulated with Leukocyte Activation Cocktail with BD GolgiPlug (BD Bioscience, Franklin Lakes, NJ, USA) according to the manufacturer’s instructions. Cells were stained with LIVE/DEAD™ Fixable Near-IR Dead Cell Stain Kit (Thermo Fisher Scientific, Waltham, MA, USA), washed, and labeled with a combination of antibodies to surface molecules: CD3, CD4, CD8, and CD19. Cells were fixed with 2% PFA (Sigma-Aldrich, St. Louis, MO, USA), permeabilized with 0.5% Saponin (Sigma-Aldrich, St. Louis, MO, USA), and stained with the following fluorescently conjugated antibodies: IL-10, IL-17, IFN-γ, and TNFα. Stained cells were acquired using LSRFortessa X20 (BD Bioscience, Franklin Lakes, NJ, USA), and data were analyzed using FlowJo software (version 10.8.51 for Mac).

### 2.5. T Cell Activation, Proliferation, and Phenotype

CD4+CD25− T cells were labeled with either 2.5 μM of CFSE or Cell Trace Violet (CTV) proliferation kit (Thermo Fisher Scientific, Waltham, MA, USA) according to the manufacturer’s instructions. 2 × 10^5^ labeled T cells were stimulated with 5 μg/mL plate-bound anti-CD3 Ab (clone OKT3, Cat. 16003785, eBioscience, San Diego, CA, USA) in combination or not with 10 μg/mL plate-bound anti-CD28 Ab (clone CD28.2, Cat. 302934, BioLegend, San Diego, CA, USA) for four days in RPMI 10% FBS (96 U-bottom well plate (VWR, Leicestershire, UK)). Different amounts of r-CL-11 [33] diluted in buffer solution (PBS CaCl_2_ 2 mM) or buffer solution alone were added to T cell culture. Following activation, T cells were stained with LIVE/DEAD Fixable Yellow Dead Cell Stain Kit (Thermo Fisher Scientific, Waltham, MA, USA), washed and labeled with a combination of antibodies to surface molecules: CD4, CD25, PD-1, HLA-DR, and CD28. For some proliferation assays, 10 or 20 μg/mL rCL-11 were pre-incubated for 30 min at room temperature with different concentrations of L-Fucose (Sigma-Aldrich, St. Louis, MO, USA) in PBS CaCl_2_ 2 mM. This sugar-blocking solution was then added to CFSE-labelled CD4+CD25− T cells activated with anti-CD3 and anti-CD28 coated antibodies for four days in RPMI 10% FBS. In other proliferation settings, CTV-labelled CD4+CD25− T cells were pre-incubated with rCL-11 (20 μg/mL) in the presence or not of different doses of RGD peptide (VWR, Leicestershire, UK) for 1 h at 37 °C and activated with plate-bound anti-CD3 and anti-CD28 antibodies for four days in RPMI 10% FBS. Stained cells were acquired using LSRFortessa X20 (BD Bioscience, Franklin Lakes, NJ, USA) and data were analyzed using FlowJo software (version 10.8.51 for Mac).

### 2.6. Confocal Microscopy

5 × 10^5^ CD4+CD25− T cells were activated or not for 48 h with anti-CD3 and anti-CD28 plate-bound antibodies (48-well plate, VWR, Leicestershire, UK) for 48 h in RPMI 10% FBS and “starved” for additional 2 h in RPMI without serum. Cells were then pre-incubated for 1 h at 37 °C with 20 µg/mL rCL-11 or buffer solution. Cells were fixed and permeabilized using the BD Cytofix/Cytoperm kit (BD Bioscience, Franklin Lakes, NJ, USA) according to the manufacturer’s instruction and blocked for 20 min at room temperature with 20% goat serum (Sigma-Aldrich, St. Louis, MO, USA). For some experiments, cells were only fixed using the Fixation Buffer (Biolegend, San Diego, CA, USA) according to the manufacturer’s instruction. Cells were then stained with primary antibodies for 1 h at 4 °C. The following antibodies were used: CL-11 (Cat. abx003772, 1: 100, Abbexa, Cambridge, UK) and CD28 (Cat. 302934, 1:100, BioLegend, San Diego, CA, USA). Cells were stained with secondary antibodies for 1 h at room temperature. Goat anti-mouse Alexa fluor 488 (Cat. A21121) and goat anti-rabbit Alexa Fluor 546 (Cat. A11010) secondary antibodies (Thermo Fisher Scientific, Waltham, MA, USA) were diluted 1:400 in 20% goat serum (Sigma-Aldrich, St. Louis, MO, USA). Cells were washed, counter-stained with DAPI (Sigma-Aldrich, St. Louis, MO, USA), and mounted on coverslips. Images were acquired in the KCL Nikon Imaging Centre by confocal fluorescence microscopy with A1R SI Confocal Microscope (×40 objective) from Nikon TM (Surrey, UK). For all samples, a cropped image of a minimum of 20 cells was used to perform the analysis. Images were analyzed using ImageJ software 1.53u.

### 2.7. Determination of CL-11 Binding to RPE and T Cells

RPE cells were treated with TrypLE Express enzyme solution (Thermo Fisher Scientific, Waltham, MA, USA) for 10 min, and single-cell suspensions were stained with LIVE/DEAD™ Fixable Near-IR Dead Cell Stain Kit (Thermo Fisher Scientific, Waltham, MA, USA). RPE cells were blocked for 20 min at room temperature with 20% goat serum. Cells were then stained with anti-CL-11 antibody (Cat. abx003772, 1:100, Abbexa, Cambridge, UK) for 30 min at 4 °C, washed and incubated with goat anti-rabbit Alexa fluor 647 secondary antibody (Cat. A21244) diluted in 20% goat serum (1:400, Thermo Fisher Scientific, Waltham, MA, USA) for 20 min at 4 °C. 5 × 10^5^ CD4+CD25− T cells were activated for 48 h with anti-CD3 and anti-CD28 plate-bound antibodies (48-well plate, VWR, Leicestershire, UK) in RPMI 10% FBS and “starved” for additional 2 h in RPMI without serum. Cells were then pre-incubated for 1 h at 37 °C with 20 µg/mL rCL-11 or buffer solution and stained as described above. Stained cells were acquired using LSRFortessa X20 (BD Bioscience, Franklin Lakes, NJ, USA), and data were analyzed using FlowJo software (version 10.8.51 for Mac).

### 2.8. Statistical Evaluation

Data are shown as mean ± standard error (SEM) unless otherwise noted. Statistical tests were prepared using GraphPad Prism software 9.5.1. Statistical significance between two experimental groups was performed using two tailed *t*-test. An RM one-way ANOVA and two-way ANOVA were used to compare one related variable between different groups. Post hoc tests were used, as indicated in the figure legends. *p* values are reported as follows: * *p* < 0.05, ** *p* < 0.01, *** *p* < 0.001, and **** *p* < 0.0001.

## 3. Results

### 3.1. Foetal-Derived RPE Cells with a Modified Immunophenotype in Response to Inflammation Inhibit T Cell Proliferation In Vitro

So far, several reports have confirmed that RPE cells, including stem cell-derived RPE cells, can function as antigen presenting cells (APCs) [12,34]. We therefore initiated the characterization of fetal RPE (fRPE) cell immunophenotype by analyzing the expression of HLA class I and class II molecules, as well the expression of co-signaling molecules, which are involved in the activation (CD80 and CD86) or inhibition (CD279, PD-L1) of T cells [35]. We used IFN-γ and hypoxia as inflammatory stimuli, as they have been shown by us and others to modify RPE cell phenotype [24,36,37]. By day 30, cultured human fRPE cells showed a cobblestone morphology and a visible pigmentation consistent with RPE maturation (Appendix A). Upon IFN-γ or hypoxia treatment, we did not observe changes in the expression of CD140b (platelet-derived growth factor receptor-β; PDGFRB), which has recently been identified as a surface marker to evaluate stem cell-RPE differentiation (Appendix A) suggesting that under these circumstances RPE cells maintain their differentiation status [38]. Flow cytometry analysis revealed that almost 100% of fRPE cells expressed HLA class I (Pan HLA), and upon IFN-γ treatment, the cells exhibited enhanced expression of HLA class I by about two-fold (MFI, mean fluorescence intensity) (Figure 1A,B). No substantial changes were observed upon hypoxia treatment. We then analyzed the expression of PD-L1 and HLA class II (HLA-DR) molecules that are known to be expressed on the surface of antigen-presenting cells [39,40]. Low expression of both PD-L1 and HLA-DR molecules was observed on the untreated fRPE cell surface, and a significant increase occurred only upon IFN-γ exposure (Figure 1A). Accordingly, the MFI of PD-L1 and HLA-DR was enhanced by about 5.8 and 2.4-fold, respectively (Figure 1B). We also detected very low expression of the costimulatory molecules CD80 and CD86 on our cell product, and their expression remained unchanged regardless of the inflammatory stimulus used. Therefore, fRPE cells acquire the potential to function as effective APCs under inflammatory conditions.

To test the function of fRPE cells, we analyzed their ability to modulate T cell responses in co-culture experiments with human peripheral blood mononuclear cells (PBMCs). Due to the ability of fRPE cells to express HLA-DR molecules following IFN-γ exposure, we focused on the CD4+ T cell subset (Appendix A) and analyzed their proliferation and cytokine production. It has been recently shown that cultured RPE cells can suppress or activate T cells in vitro [41]. Therefore, we first co-cultured fRPE cells with PBMCs in the absence of other exogenous stimuli to test their ability to promote a T cell response. Our data showed no proliferation either in the control sample (PBMCs without RPE cells) or following co-culture with RPE cells treated, as indicated in Appendix A. As shown in Figure 1C (left column), CD4+ T subsets stimulated by the addition of anti-CD3 agonist antibody were inhibited when allogeneic fRPE cells were added to the culture, irrespective of the pre-conditioning conditions. Such abrogation of T cell proliferation was also observed within the CD8+ pool (Appendix A). Accordingly, compared with control cultures (T cells without fRPE cells), the percentage of CD4+ T cells expressing pro-inflammatory cytokines, namely IL-17, IFN-γ, and TNF⍺ [42] (Figure 1C,D) decreased when co-cultured with fRPE cells. The frequency of CD4+ T cells expressing IL-10, a cytokine with anti-inflammatory properties [43], was very low and did not increase following RPE cell co-culture (Figure 1C,D). These observations suggest the ability of in vitro cultured fRPE cells to downmodulate CD4+ T cell alloimmune response. To further confirm the immunosuppressive properties of fRPE cells, we repeated the co-culture assay by using purified CD4+CD25− T cells to rule out the possibility of a suppressive effect mediated by regulatory T cells (CD4+CD25+), which are present in the CD4+ T cell pool. As shown in Appendix A, CD4+CD25− T cells were unable to proliferate when co-cultured with fRPE cells (first row) and inhibited when the soluble anti-CD3 antibody was added to the culture (second row). In addition, the presence of allogeneic dendritic cells (DCs) could not induce T cell proliferation when co-cultured with fRPE cells confirming the ability of these cells to suppress the activation of DCs and therefore prevent T cell activation [44,45]. During the co-culture, PBMCs also promoted a further increase of HLA class I, HLA-DR, and PD-L1 expression in fRPE cells in addition to the inflammatory pre-treatment, thus potentiating these cells to interact with allogeneic T cells (Figure 1E). These results confirm the immunosuppressive phenotype of in vitro cultured fRPE cells and their ability to express or up-regulate specific surface markers according to the inflammatory conditions.

### 3.2. CL-11 Produced by Inflammatory fRPE Cells Can Function as a Modulator of CD4+CD25− T Cell Activation In Vitro

Our recent work has shown that under stress conditions, stem cell-derived RPE cells are able to secrete and bind the complement component CL-11, which activates the lectin pathway of the complement cascade [24,26]. Besides its canonical role as a trigger of complement activation against invasive pathogens [27,46], CL-11 and other collectins, such as MBL and C1q, have been shown to inhibit T cell immune responses [47]. As fRPE cells switched to an immunogenic phenotype following IFN-γ exposure (Figure 1A), we questioned whether this pro-inflammatory RPE phenotype could induce the binding of CL-11 to the fRPE cell surface and whether RPE-secreted CL-11 could modulate CD4+CD25− T cell activation. We cultured fRPE cells under the inflammatory conditions described above, and we analyzed the surface binding of CL-11 to the RPE. Because serum was absent in our culture conditions, the only source of CL-11 was through RPE cell production [24]. Our results detected significant CL-11 product bound to fRPE cells after IFN-γ treatment, besides confirming the already known effect of hypoxia on CL-11 binding to RPE cells (Figure 2A,B) [24]. After this initial RPE pre-conditioning, the percentage of CL-11+ RPE cells tended to decrease over time (Figure 2C), which could represent a re-distribution between the pool of membrane-bound, internalized, or dissociated CL-11 molecules. Nonetheless, our data confirm the ability of RPE cells to secrete CL-11, as shown by enhanced cell-autonomous binding in a pro-inflammatory context [24,30].

To define the effect of soluble CL-11 on T cell immune response, we activated CD4+CD25− T cells with plate-bound αCD3 or αCD3/αCD28 in the presence or not of added soluble hrCL-11. By using αCD3 or αCD3/αCD28 stimulation, we aimed to mimic the classic two signals for T cell activation, namely ligation by TCR and MHC-peptide complex in combination or not with the engagement of CD28. We ruled out using a combination of different antibodies (e.g., αCD3/αPD-L1) as they can induce a different program in CD4+ T cells than αCD3/αCD28 co-stimulation resulting in the generation of Treg cells (iTreg cells) [48]. We then analyzed the expression of the activation markers CD25 (IL2Rα) and HLA-DR and the expression of co-signaling molecules, CD28 and PD-1, which respectively mediate activation and inhibition of T cell response. After four days of culture, the percentage of CD25+ T cells was significantly lower in the presence of a high concentration of rCL-11 (20 µg/mL) (mean of 29.6%) following stimulation with αCD3/αCD28 compared to untreated T cells or those cultured with 1 or 5 µg/mL of rCL-11 (mean of 51%, 57%, and 52.4%, respectively). High concentrations of rCL-11 also promoted suppression of HLA-DR expression, consistent with an inhibitory effect of CL-11 on CD4+ T cell activation. Of note, we observed dramatic downregulation of CD28 molecules on T cells cultured with 20 µg/mL of rCL-11 and an increase of PD-1+ cells under the same conditions (Figure 2D,E). The lack of downregulation of CD28 expression in the untreated control activated with αCD3/αCD28 ruled out the possibility of a downmodulation induced by the activation process itself [49]. The presence of rCL-11 in conjunction with CD3 crosslinking was insufficient to induce changes in the detection of the markers listed above (Figure 2E) which occurred only in the presence of co-stimulation. These data suggest that the putative receptor for CL-11 on T cells does not directly inhibit the TCR signal and rather may attenuate CD28-costimulation by downmodulating the surface expression of CD28 molecules.

### 3.3. The Binding of CL-11 to T Cells Results in Increased Intracellular CD28 Detection

To further address the proposed interaction of CL-11 with T cells resulting in suppression of CD28 expression, we performed confocal microscopy on CD4+CD25− T cells activated for 48 h with αCD3/αCD28 and pre-incubated with or without rCL-11. As shown in Figure 3A,B, the binding of CL-11 to T cells resulted in a decrease of the total CD28 signal intensity (IntDensity) compared to the untreated group. Such decrease of CD28 signal also occurred in the unstimulated group, although to a lesser extent (Appendix A), suggesting that the activation by αCD3/αCD28 per se is necessary but not sufficient to initiate this process. The decrease of CD28 intensity was even more pronounced by analyzing the CD28 intensity per cell. For this analysis, we sub-selected those cells with positive staining for CL-11 (Figure 3C, yellow arrows) or with low or negative staining for CL-11 (Figure 3C, violet arrows). As shown in Figure 3D, the mean intensity of the CD28 signal was dramatically lower on CL-11 positive cells (CL-11+) compared to the negative group (CL-11-), confirming that the binding of CL-11 triggered CD28 downmodulation.

However, in those permeabilized cells where the detection of CL-11 was particularly strong, we observed a concomitant increase of CD28 signal (Figure 3E, lower panels) compared to the untreated group (Figure 3E, upper panels). Confocal microscopy combined with analysis of the protein colocalization coefficient suggested that CL-11 and CD28 reside partly in overlapping locations in both resting and activated T cells (Appendix A). Due to the permeabilization step in our protocol, we could not distinguish surface versus intracellular staining of CL-11 and CD28. Therefore, to better resolve the relationship between these two molecules, we also examined non-permeabilized cells by confocal microscopy. As shown in Appendix A, the mean intensity of the surface CD28 signal was significantly reduced in the rCL-11 treated group, confirming our previous observation with FACS analysis (Figure 2D,E). Moreover, following pre-incubation with rCL-11, the intracellular CD28 mean fluorescence was clearly increased compared to surface staining of CD28 alongside CL-11 (Appendix A). Thus, CL-11 appeared to induce a significant increase of cytoplasmic CD28 (Figure 3F) alongside CL-11 (Figure 3G). Altogether these confocal microscopy results suggest that CL-11 interaction with the T cell enhanced the internalization of CD28 from the cell surface, though the results do not distinguish this from the possibility of CL-11 impeding the translocation of CD28 from within the cell to the cell surface.

### 3.4. The Inhibition of T Cell Proliferation Was Mediated by the Binding of CL-11 through Its CLD Domain

Having shown that CL-11 can modulate the immune phenotype of CD4+CD25− T cells, we investigated the direct effect of rCL-11 on the T cell proliferative response in a dose-dependent manner. We found a marked inhibition of proliferation of αCD3/αCD28 activated CD4+CD25− T cells cultured for four days in the presence of the higher concentration of soluble rCL-11 (Figure 4A,B).

We previously demonstrated that CL-11 binds to stressed epithelial cells through its CRD domain, inducing complement activation via the lectin pathway [24,29]. Such sugar-binding activity, especially to L-fucose and D-mannose motifs, accounts for its antibacterial activity in host defense [26,50]. To assess whether binding of CL-11 to the T cell surface occurred through its CRD domain, we pre-incubated rCL-11 (10 and 20 µg/mL) with an increasing amount of L-fucose to block the CRD of rCL-11, prior to measuring the effect of rCL-11 on T cell proliferation [51]. As shown in Figure 4C, neither the binding of CL-11 to T cells nor the anti-proliferative effect of CL-11 on T cells was reversed by the addition of L-fucose, indicating that the CRD domain was not responsible for the observed interaction of CL-11 with the T cell preparation. Having found no evidence that the binding of CL-11 to T cells was sugar-blockable, we considered whether the binding to T cells used the CLD (not the CRD) of CL-11. For example, it has been proposed that calreticulin exposed on T cells can act as a receptor interacting with CLD of the collectins MBL and CL-11 [47]. To elucidate the possible role of a T cell collagen-binding receptor for CL-11, we examined whether the T cell/CL-11 interaction was inhibitable with RGD peptide. RGD peptide is a synthetic compound made up of the arginine–glycine–aspartate motif that has been extensively used to block receptor/collagen–ligand interactions [52]. Here we incubated CD4+CD25− T cells with rCL-11 in the presence of RGD peptide and then measured the proliferative response to T cell activation with αCD3/αCD28. We found that the presence of RGD partially restored the T cell proliferative response induced by αCD3/αCD28 activation (Figure 4D,E); it also led to reduced T cell binding by CL-11 (Figure 4F) consistent with the ability of RGD peptide to disrupt the interaction of CL-11 with a receptor expressed on T cells. Overall, our data indicate that T cell binding by CL-11 occurred through the CLD domain of CL-11, urging further interest in the nature of CL-11 ligand on T cells.

## 4. Discussion

CL-11 is a member of the lectin family of pattern recognition molecules, with known antimicrobial functions and the ability to activate the lectin pathway of the complement cascade after binding to the target surface through its carbohydrate recognition domain (CRD) thereby promoting inflammation and enhancement of T cell priming [25,53]. In this study, we show a novel counterintuitive effect of CL-11 which resulted in a dose-dependent anti-proliferative effect on CD4+CD25− T cells associated with surface CD28 downmodulation and increase of its intracellular localization.

In agreement with recently published findings by Zhao and co-authors [47], our data show that CL-11 can regulate the T cell response through the binding of the collagen-like domain (CLD) of CL-11 to helper T cells. Our findings, however, stand out for two key aspects. Firstly, the binding of CL-11 to the T cell surface promoted a notable loss of CD28 cell surface expression together with enhanced intracellular localization of CD28 molecules. Secondly, the rescue of T cell proliferation occurred in the presence of RGD peptide, suggesting that T cell binding by CL-11 occurred through a distinct collagen-binding mechanism. One of the most characterized functions of CD28 is the ability to organize the cytoskeleton, which is crucial for the dynamic and functional assembly of the immunological synapse (IS). Indeed, the actin cytoskeleton rearrangements allow the recruitment and trapping of membrane rafts in the IS to sustain and amplify TCR signaling in the presence of CD28 co-stimulation, both of which are necessary for T cell activation. In addition, the evidence that CD28 binds to F-actin, an actin-binding protein, sustains the hypothesis that actin-driven forces could be involved in the localization of CD28 [54,55,56]. These published data, together with our finding that the binding of CL-11 resulted in predominant intracellular detection of CD28, suggest that the interaction between the CLD domain of CL-11 and its putative collagen-receptor expressed on T cells could interfere with the actin cytoskeleton arrangements and, accordingly, CD28 localization. This could result either in a lack of CD28 translocation to lipid rafts [57] or enhanced recruitment of CD28 to the endocytic machinery [58]. In addition, the partial rescue of T cell proliferation in the presence of RGD peptide persuades us to speculate that blockade of the interaction between CL-11 and its putative collagen-like receptor expressed on T cells could restore actin cytoskeleton dynamics and, ultimately, T cell activation and proliferation. Due to the limited dose range of RGD peptide used in the study alongside the partial effect observed and the lack of an in vivo model to test our hypothesis, future studies are needed to identify this putative ligand and its functional interactions.

The concept of the complement-independent function of lectin pattern-recognition molecules (PRMs) involved in the regulation of effector T cell response is not new. Indeed, the immunomodulatory properties mediated by serum collectins were initially confined to surfactant proteins, SP-A and SP-D [59], and more recently, they have been extended to MBL, C1q, and CL-11 [47]. Such non-complement function is known to be mediated by the interaction of the CLD with Calreticulin/CD91, which, according to cell type, can mediate distinctive biological functions. For example, the interaction between CL-11 and calreticulin (CRT)/CD91 on RPE cells mediated cell phagocytosis [30], whereas such interaction on proximal tubular epithelial cells (PTECs) resulted in p38 MAPK signaling pathway activation and up-regulation of inflammatory mediators [60]. Given this background, along with our observations, it is now important to understand the entity of the dual biological activity of collectins in order to facilitate the appropriate therapeutic design.

We initiated this study to address whether CL-11 expression by RPE cells could have a negative impact on their potential use for stem cell transplant therapy. The demonstration here of an immunoinhibitory action CL-11 versus its reported pro-inflammatory effect on RPE cells brings the question of the duality of function to the forefront. Our previous identification of human RPE cells as a source of endogenous CL-11 up-regulated under hypoxic conditions [24], together with other reports of inflammatory stress through endogenous production of complement [19,61,62], led us to speculate that local production of CL-11 (under inflammatory conditions) in the retina might facilitate a rapid response by infiltrating lymphocytes [21,63,64,65]. To reconcile the pro-inflammatory and immunosuppressive actions of CL-11, we suggest that tissue response might be dose- and time-dependent. We propose that low-dose acute hypoxia may trigger the inflammatory RPE phenotype mitigating against RPE transplant acceptance, whereas high-dose production of CL-11 and T cell binding by a putative collagen receptor could mediate adaptive immune suppression. In the eye, other cell sources, in addition to RPE cells, such as microglia and photoreceptors, would be expected to contribute to the pool of soluble CL-11, perhaps sustaining RPE transplant acceptance or mediated low-grade chronic rejection (Figure 5).

In summary, our study showed that CL-11 exerts an immunomodulatory action on the T cell response mediated by the interaction of its CLD with an undefined T cell ligand. Because this phenomenon occurred at high CL-11 concentration, we speculate that low avidity binding of CL-11 through its CLD could favor a complement-independent homeostatic role relevant to the immune-privileged status of the eye. In addition, our data clearly demonstrate that the immunosuppressive effect was associated with lower surface expression and increased intracellular location of CD28, confirmed by flow cytometry and confocal microscopy.

## 5. Conclusions

Collectin-11 is best known as a positive regulator of immunity particularly through activating complement on tissues such as the retinal epithelium. Potentially, this impedes the use of RPE for transplantation. Counterintuitively, we show a negative regulatory effect of collectin-11 on T cell activation, which may offset the pro-inflammatory impact on the immune response. Due to its dual biological activity, collectin-11 should be regarded with increased appreciation. Continuing investigations of the mechanisms used by RPE cells to orchestrate the alloimmune response under inflammatory conditions will pave the way for novel therapeutic strategies to reduce the inflammatory response following RPE transplantation.

## Figures and Tables

**Figure 1 cells-12-01805-f001:**
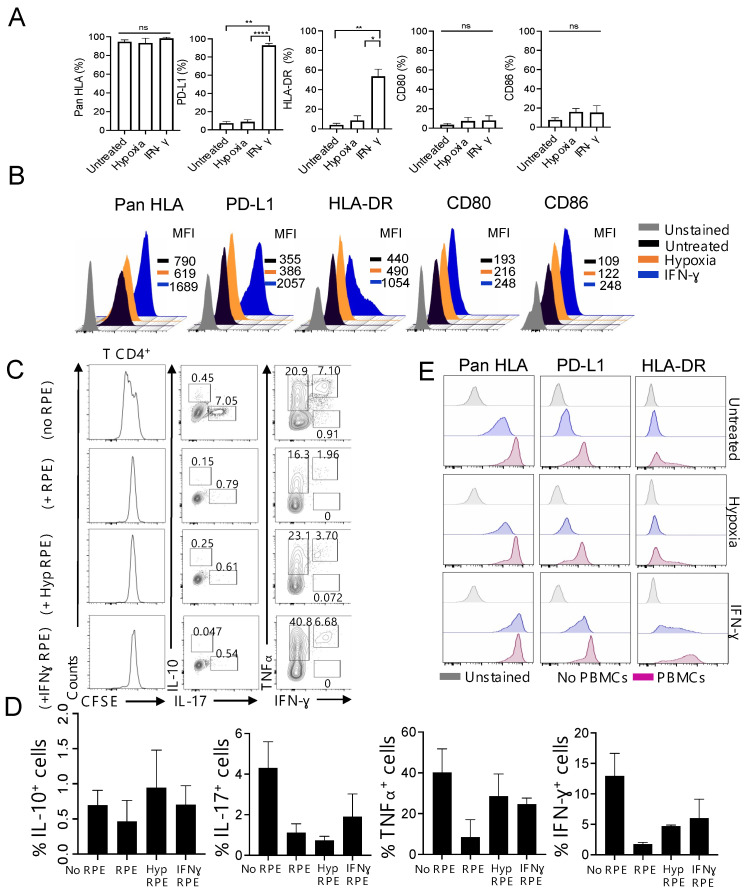
Immunosuppressive properties of in vitro cultured fRPE cells. (**A**) fRPE cells were treated for 24 h under hypoxic conditions or for three days with 500 IU/mL of hrIFN-γ. The expression of Pan HLA, PD-L1, HLA-DR, CD80, and CD86 was analyzed by Flow Cytometry and quantified. Data are expressed as mean ± SEM and are pooled from four different RPE cell lines. * *p* < 0.05, ** *p* < 0.01, and **** *p* < 0.0001 were considered significant using One-way ANOVA followed by Tukey multiple comparisons. (**B**) Representative flow cytometry histograms showing the mean fluorescence intensity (MFI) of the above surface markers under the indicated conditions. (**C**) Representative histograms showing CFSE dilution of PBMCs (5 × 10^5^), gated of CD4+ T cells, activated with soluble αCD3 (0.5 µg/mL) for four days in the presence or not of 5 × 10^5^ fRPE cells pre-treated or not as indicated (left column). Representative dot plots showing the expression of the indicated cytokines by gated CD4+ T cells upon co-culture with pre-treated or not RPE cells (middle, right column). Histograms and dot plots are representative of two independent experiments. (**D**) Quantification of cytokine production by gated CD4+ T cells under the indicated conditions. Data are expressed as mean ± SEM and are pooled from at least two independent experiments. (**E**) Representative flow cytometry histograms showing the mean fluorescence intensity (MFI) of Pan HLA, PD-L1, and HLA-DR on RPE cells under the indicated conditions co-cultured or not with PBMCs (5 × 10^5^) activated with soluble αCD3 (0.5 µg/mL) for four days.

**Figure 2 cells-12-01805-f002:**
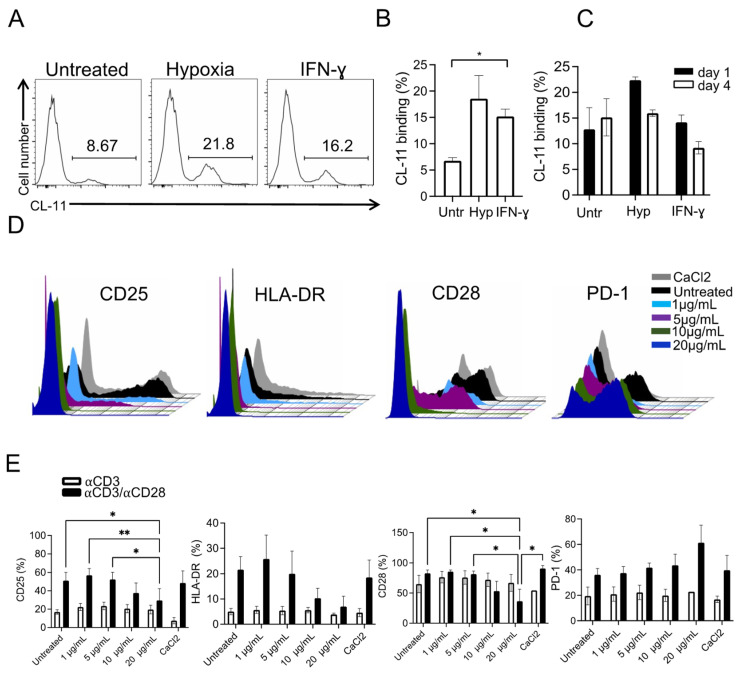
Inhibition of CD4+CD25− T cell activation triggered by CL-11. (**A**) Representative flow cytometry histograms showing surface binding of CL-11 to fRPE cells treated for 24 h under hypoxic conditions or for three days with 500 IU/mL of hrIFN-γ. (**B**) Quantification of CL-11 binding to fRPE treated as in (**A**) and following additional four days in culture at 37 °C (**C**). Data are expressed as mean ± SEM and are pooled from three (**A**) or two (**C**) different fRPE cell lines. * *p* < 0.05 was considered significant using One-way ANOVA followed by Tukey multiple comparisons. (**D**) Representative flow cytometry histograms showing CD25, HLA-DR, CD28, and PD-1 expression on CD4+CD25− T cells activated with αCD3/αCD28 for four days in the presence or not of different concentrations of rCL-11 or buffer solution (CaCl2). (**E**) Quantification of the surface molecules listed above on CD4+CD25− T cells activated with αCD3 or αCD3/αCD28 for four days in the presence or not of different concentrations of rCL-11 or buffer solution (CaCl_2_). Data are expressed as mean ± SEM and are pooled from four different donors. * *p* < 0.05 and ** *p* < 0.01 were considered significant using One-way ANOVA followed by Tukey multiple comparisons.

**Figure 3 cells-12-01805-f003:**
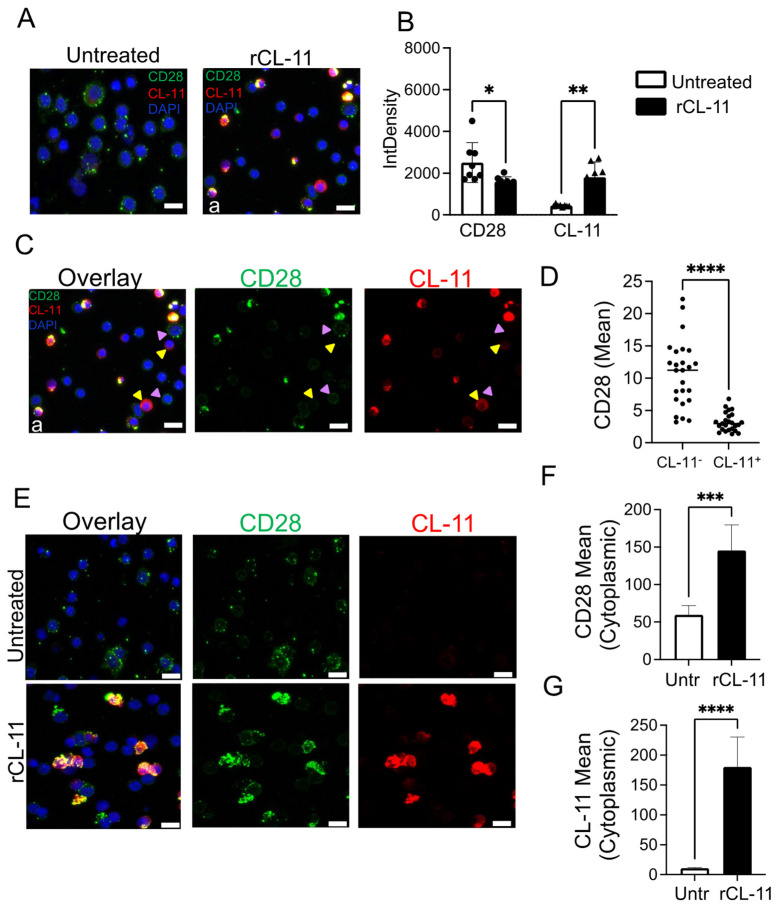
CL-11 promotes the intracellular localization of CD28 molecules. (**A**) Representative high magnification images showing CD28 and CL-11 expression by CD4+CD25− T cells activated with αCD3/αCD28 for 48 h and pre-incubated for 1 h at at 37 °C with buffer solution (untreated) or rCL-11. CD28 (green), CL-11 (red), and nuclei (blue) are shown. Scale bars represent 25 µM. See also Appendix A (uncropped images). (**B**) Quantification of integrated density for CD28 (bullet point) and CL-11 (triangle) fluorescence using ImageJ software. Data are expressed as mean ± SEM. Eight cropped images per condition of a minimum of 20 cells are pooled from three different healthy donors. * *p* < 0.05 and ** *p* < 0.01 were considered significant using two-way ANOVA followed by Šídák multiple comparisons. (**C**) High magnification confocal microscopy image (a) of A showing CD28 (green), CL-11 (red), and Nuclei (blue) overlay and CD28 (green) and CL-11 (red) single channels. Scale bars represent 25 µM. (**D**) Quantification of CD28 mean fluorescence using ImageJ software. Data are expressed as individual values (25) per condition (CL-11+ and CL-11-). Five cropped images from rCL-11 treated group are pooled from at least two different donors. **** *p* < 0.0001 was considered significant using two-tailed *t*-test. (**E**) High magnification confocal microscopy image showing CD28 (green), CL-11 (red), and Nuclei (blue) overlay and CD28 (green) and CL-11 (red) single channels on CD4+CD25− T cells activated with αCD3/αCD28 for 48 h and pre-incubated for 1 h at at 37 °C with buffer solution (untreated) or rCL-11. Scale bars represent 25 µM. See also Appendix A (uncropped images). (**F**) Quantification of cytoplasmic CD28 mean fluorescence using ImageJ software. Data are expressed as mean ± SEM. Five cropped images for the untreated group and seven cropped images for the rCL-11 treated group of a minimum of 20 cells are pooled from three different healthy donors. *** *p* < 0.001 was considered significant using two-tailed *t*-test. (**G**) Quantification of cytoplasmic CL-11 mean fluorescence using ImageJ software. Data are expressed as mean ± SEM. Five cropped images for the untreated group and seven cropped images for the rCL-11-treated group of a minimum of 20 cells are pooled from three different healthy donors. **** *p* < 0.0001 was considered significant using two-tailed *t*-test.

**Figure 4 cells-12-01805-f004:**
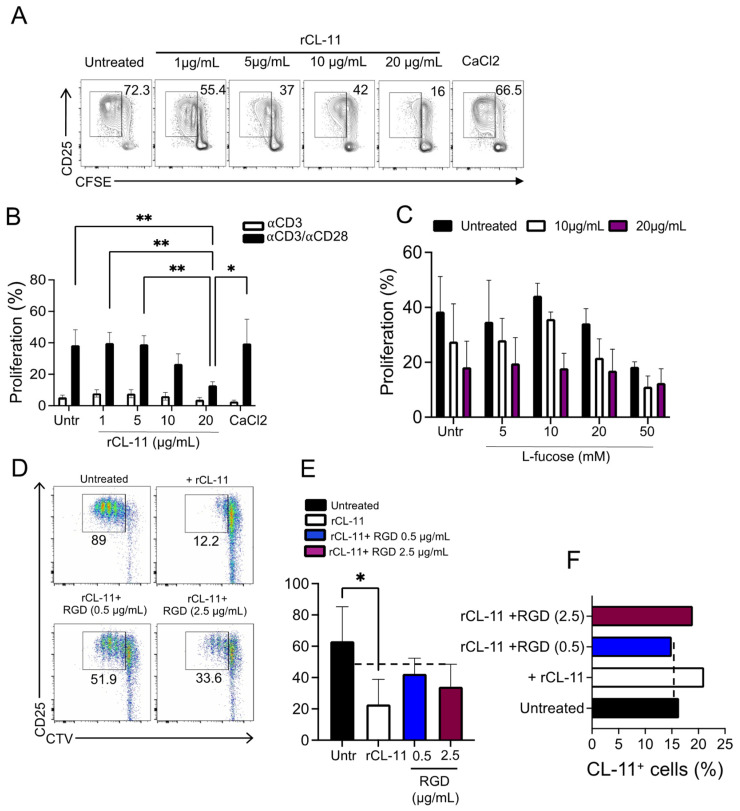
CL-11 uses its CLD domain to bind and inhibit T cell proliferation. (**A**) Representative plots showing proliferation (CFSE dye dilution) and CD25 expression of CD4+CD25− T cells stimulated with αCD3/αCD28 for four days in the presence or not of different amounts of rCL-11 or buffer solution (CaCl_2_). (**B**) Proliferation (CFSE dye dilution) of T cells activated for four days with αCD3 (*n* = three different donors) or αCD3/αCD28 (*n* = five different donors) under the culture conditions described in A. Data are expressed as mean ± SEM. * *p* < 0.05 and ** *p* < 0.01 were considered significant using two-way ANOVA followed by Tukey multiple comparisons. (**C**) Proliferation (CFSE dye dilution) of T cells pre-incubated with different concentrations of soluble L-fucose in the presence or not of 10 µg/mL or 20 µg/mL of rCL-11. Data are expressed as mean ± SEM and are pooled from two different donors. (**D**) Representative plots showing proliferation (CTV dye dilution) and CD25 expression of CD4+CD25− T cells in the presence or not of 20 µg/mL of rCL-11 (first row). T cells were pre-incubated with 20 µg/mL of rCL-11 in the presence of 0.5 µg/mL or 2.5 µg/mL of RGD peptide (second row). CD4+CD25− T cells were then stimulated with αCD3/αCD28 for four days. (**E**) Quantification of T cell proliferation shown in (**D**). Data are expressed as mean ± SEM and are pooled from four different donors. * *p* < 0.05 was considered significant using One-way ANOVA followed by Tukey multiple comparisons. (**F**) Representative graph showing the percentage of cells bound by CL-11 (CL-11+ cells) treated as in (**D**).

**Figure 5 cells-12-01805-f005:**
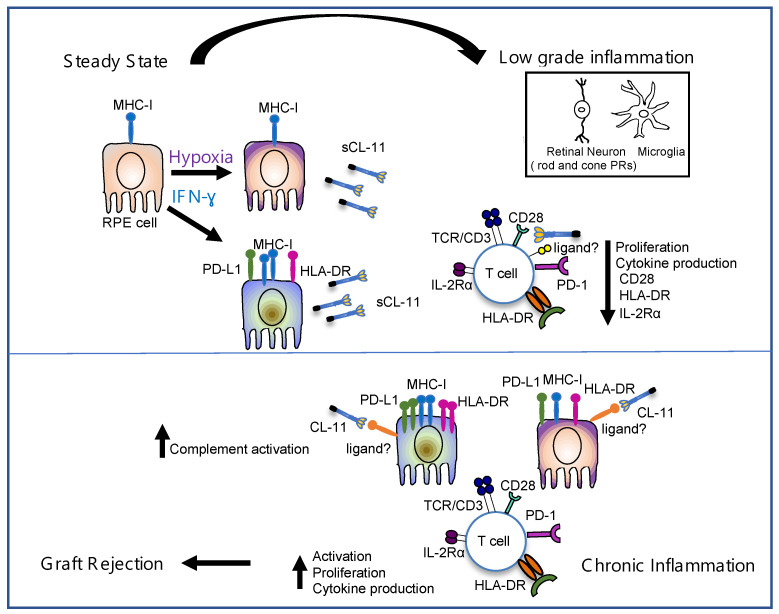
Proposed scheme by which locally produced CL-11 regulates the immune response of infiltrating effector T cells following RPE transplantation. Under normal physiological conditions (steady state), RPE cells constitutively express MHC class I molecules. Following inflammatory stimuli (e.g., hypoxia, IFN-γ), the RPE cells become immunogenic and can express different levels of MHC class I and MHC class II molecules and PD-L1 according to the nature and intensity of the stimulus. T cell activation is either promoted or inhibited depending on the balance of activating (i.e., MHC class II) and inhibitory (i.e., PD-L1) signals and soluble molecules. CL-11 is produced and secreted by stressed RPE cells and can contribute to maintain retinal homeostasis by inhibiting the T cell alloresponse at a low level of inflammation (low-grade inflammation). Other cell sources, such as microglia and photoreceptors, are believed to contribute to the pool of soluble factors that exhibit immunosuppressive properties. In contrast, under chronic inflammatory conditions, we propose that CL-11 may favor immunogenicity through enhanced complement activation and T cell stimulation, contributing to RPE graft rejection. MHC-I: major histocompatibility complex (MHC) class I; MHC-II: major histocompatibility complex (MHC) class II; PD-L1: programmed cell death 1 ligand 1; PD-1: programmed cell death 1; IL2Rα (CD25): α chain of the high-affinity IL-2 receptor; CL-11: Collectin-11.

## Data Availability

The data that support the findings of this study are available from the corresponding author, [G.F.], upon reasonable request.

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
