# Peer review of "Soluble Collectin 11 (CL-11) Acts as an Immunosuppressive Molecule Potentially Used by Stem Cell-Derived Retinal Epithelial Cells to Modulate T Cell Response"

_cells, 2023, doi:10.3390/cells12131805_

Round 1
Reviewer 1 Report
The authors are evaluating the role of Collectin 11 (CL-11) on T- cell immune response. The study suggests that CL-11 may have a complement-independent immunoregulatory function on immune cells, potentially contributing to the down-regulation of the T cell response in the context of retinal pigment epithelium transplantation. Even though the authors are addressing a very relevant question, the following comments need to be addressed to enhance the manuscript further.
Comments
1. Numerous references have different configuration styles within the manuscript. It would be important to maintain consistency in the formatting.
2. Please provide catalog numbers of the antibodies used for the study.
3. Within lines 210-218, the authors mention the fold increase of HLA class 1 and class II. Does PAN HLA stand for HLA class 1, if yes, please mention it.
4. The authors need to clarify the selection of the various time frames for different treatment.
5. Remove hyphen from the words “polarized”, “participating” in the introductory section line no. 31 and 52 respectively.
6. Italicize the word in vitro.
7. The sentences within lines 74-78, 196-199 and 236- 242 are too complex and can be reworded or broken down to multiple sentences.
8. Remove the additional period between lines 199-210.
Author Response
We would like to thank the reviewer for the positive comments. We have answered all the questions raised below.
- Numerous references have different configuration styles within the manuscript. It would be important to maintain consistency in the formatting.
We have carefully checked the formatting of the references which are now all consistent with the MDPI style as required by the journal.
- Please provide catalog numbers of the antibodies used for the study.
The full list of Catalog. Numbers of labelled antibodies used for Flow Cytometry has been provided in the Table 1 (Supplementary Materials). The missing Cat. Numbers for unlabelled or secondary antibodies used in this study have now been added in the section of Material and Methods and highlighted in yellow.
- Within lines 210-218, the authors mention the fold increase of HLA class 1 and class II. Does PAN HLA stand for HLA class 1, if yes, please mention it.
We appreciate this suggestion and have now clarified in the text that Pan HLA stands for HLA class I as follows: “Flowcytometry analysis revealed that almost 100% of fRPE cells expressed HLA class I (Pan HLA)..”( line 224)
- The authors need to clarify the selection of the various time frames for different treatment.
We appreciate this suggestion and have now added an additional reference (line 95, ref 31) to justify the selection of the various time frames.
- Remove hyphen from the words “polarized”, “participating” in the introductory section line no. 31 and 52 respectively.
The hyphen has been removed and the correction has been highlighted in yellow
- Italicize the word in vitro.
The word in vitro has been italicized as requested by the reviewer and highlighted in yellow
- The sentences within lines 74-78, 196-199 and 236- 242 are too complex and can be reworded or broken down to multiple sentences.
We have revised the narrative to improve the flow and readability. All the changes are highlighted in yellow in the revised manuscript for ease of reference. The sentences modified are now within lines 74-78, 209-212 and 248-254 in the revised manuscript.
- Remove the additional period between lines 199-210.
The additional period has been removed as required (line 223)
Reviewer 2 Report
The purpose of this study was to investigate the role of collectin-11 (CL-11, involved in the complement system and binding to T-cells) on T-cell immuno responses. RPE cells co-cultured with T-cells inhibited the proliferation of T-cells. However when inflammation was induced by pre-treatment of RPE cells with IFN-gamma, hypoxia, or coculture with dendritic cells, RPE cells upregulated MHC class I and expressed MHC class II. T-cell binding of CL-11 inhibited T-cell proliferation. The authors demonstrate that binding of CL-11 to T-cells occurs through its CLD domain, and that the peptide RGD can abolish the inhibitory effect of CL-11 on T-cell proliferation. . RPE cells can bind and secrete CL-11 under stress responses. CL-11 thus has dual functions as a trigger of the complement cascade and as modulator of T-cell responses. The results of this study are relevant for RPE cell allotransplantation.
General comments:
This is a well-written paper. The figures are good. Especially good is the summary diagram (figure 5). The reviewer has only minor comments.
Specific comments:
Materials and Methods: please add City and state to the information about suppliers when first mentioned.
There is some variation in the number of leucocyte donors (n = 2-5) used for different experiments. The authors should explain why.
Results:
(minor) p.11, line 431: “Figure 4D and Figure E” should be “Figure 4D,E”
Typo in supplementary materials: “Supplemetary”
Author Response
We would like to thank the reviewer for the positive comments. We have answered all the questions raised below.
1- Materials and Methods: please add City and state to the information about suppliers when first mentioned.
City and State of the Suppliers have now been added and highlighted in yellow.
2-There is some variation in the number of leucocyte donors (n = 2-5) used for different experiments. The authors should explain why.
We thank the reviewer for allowing us to clarify the reasons behind the variation of the numbers of donors used for different experiment. The variation in the number was linked to the nature of the experiment performed. For example, in the co-culture assays, the number of fRPE cells available combined with the different culture conditions used (Hypoxia and IFNg) did not allow us to use more than 2 donors. In other experimental settings, we could instead use a higher numbers of donors
Results:
3- (minor) p.11, line 431: “Figure 4D and Figure E” should be “Figure 4D,E”
We have now modified “Figure 4D and Figure E” into “Figure 4D- E” ( line 440 in the revised manuscript)
4- Typo in supplementary materials: “Supplemetary”
We have now corrected the typo.
Reviewer 3 Report
Retinal pigment epithelium (RPE) cell allotransplantation is a potential solution for various retinal diseases. However, the activation of the RPE-complement system through collectin-11 (CL-11) binding poses a barrier to successful transplantation, as it triggers a complement-mediated inflammatory response that promotes T cell recognition. To address this, researchers investigated the role of CL-11 in T cell immune response. They found that RPE cells, in an inflammatory environment, upregulated MHC class I and II molecules and inhibited T cell proliferation when co-cultured. CL-11 was identified as partially responsible for this effect, as its binding to T cells reduced proliferation and downregulated CD28 expression. The suppressive action of CL-11 was reversed when T cell binding was blocked by an RGD peptide. It was observed that RPE cells can secrete CL-11 under stress, suggesting soluble CL-11 contributes to the immunosuppressive properties of RPE cells. This investigation sheds light on the dual biological activity of CL-11 as a complement cascade trigger and a modulator of T cell responses, providing insights into the immunogenic properties of RPE cells.
The authors presented a very interesting with addecuate methodology research work. Despite not having any suggestions or critizism about the work presented, I do miss some experiment or results more functional, either in vivo or in vitro work.
Apart from that, my only comment is that the figure 5, the quality could be improved.
Congratulations for such a nice work.
Author Response
We would like to thank the reviewer for the positive comments.